# Transcriptomic modulation in response to an intoxication with deltamethrin in a population of *Triatoma infestans* with low resistance to pyrethroids

**Lucila Traverso[1], Jose Manuel Latorre Estivalis[2], Gabriel da Rocha Fernandes[3], Georgina Fronza[4], Patricia Lobbia[5], Gastón Mougabure Cueto[5], Sheila Ons**[1] *

**1** Laboratorio de Neurobiología de Insectos (LNI), Centro Regional de Estudios Genómicos, Facultad de Ciencias Exactas, Universidad Nacional de La Plata, CENEXA, CONICET, La Plata, Buenos Aires, Argentina, **2** Laboratorio de Insectos Sociales, Instituto de Fisiología, Biología Molecular y Neurociencias, Universidad de Buenos Aires—CONICET, Ciudad Autónoma de Buenos Aires, Argentina, **3** Plataforma de Bioinformática, Instituto René Rachou—FIOCRUZ, Belo Horizonte, Minas Gerais, Brazil, **4** Laboratorio de Ecología de Enfermedades Transmitidas por Vectores, Instituto de Investigación e Ingeniería Ambiental, Universidad Nacional de San Martín—CONICET, San Martín, Buenos Aires, Argentina, **5** Laboratorio de Investigación en Triatominos (LIT), Centro de Referencia de Vectores (CeReVe), Ministerio de Salud de la Nación, CONICET, Santa María de Punilla, Córdoba, Argentina

* sheila.ons@presi.unlp.edu.ar, sheilaons@gmail.com

**Data Availability Statement:** All raw data files are available from the NCBI SRA database (accession

## Abstract

### Background

*Triatoma infestans* is the main vector of Chagas disease in the Southern Cone. The resistance to pyrethroid insecticides developed by populations of this species impairs the effectiveness of vector control campaigns in wide regions of Argentina. The study of the global transcriptomic response to pyrethroid insecticides is important to deepen the knowledge about detoxification in triatomines.

### Methodology and findings

We used RNA-Seq to explore the early transcriptomic response after intoxication with deltamethrin in a population of *T. infestans* which presents low resistance to pyrethroids. We were able to assemble a complete transcriptome of this vector and found evidence of differentially expressed genes belonging to diverse families such as chemosensory and odorant-binding proteins, ABC transporters and heat-shock proteins. Moreover, genes related to transcription and translation, energetic metabolism and cuticle rearrangements were also modulated. Finally, we characterized the repertoire of previously uncharacterized detoxification-related gene families in *T. infestans* and *Rhodnius prolixus*.

### Conclusions and significance

Our work contributes to the understanding of the detoxification response in vectors of Chagas disease. Given the absence of an annotated genome from *T. infestans*, the analysis presented here constitutes a resource for molecular and physiological studies in this

number Bioproject PRJNA778244 - https://www.ncbi.nlm.nih.gov/bioproject/?term=PRJNA778244).

**Funding:** This work has been supported by Argentinean Agencia Nacional de Promoción de la Innovación, el Desarrollo Tecnológico y la Innovación http://www.agencia.mincyt.gob.ar/; PICTstartup 2018-0275 and PICT2018-0862 to S. O., PICT-2019-01533 to LT. The funders had no role in study design, data collection and analysis, decision to publish, or preparation of the manuscript.

**Competing interests:** I have read the journal's policy and the authors of this manuscript have the following competing interests: J.M.L.E, G.F, P.L, G. M.C and S.O are investigators from Consejo Nacional de Ciencia y Tecnologia (CONICET; https://www.conicet.gov.ar/). L.T. is recipient of a research fellowship from CONICET.

species. The results increase the knowledge on detoxification processes in vectors of Chagas disease, and provide relevant information to explore undescribed potential insecticide resistance mechanisms in populations of these insects.

## Author summary

Chagas disease affects millions of people worldwide. In the Southern Cone, the development of pyrethroid resistant populations from *T. infestans* is related to vector persistence and affects the efficiency of vector control campaigns. Several studies have explored the causes of insecticide resistance in *T. infestans* populations. However, the global transcriptomic response after insecticide treatment has not been analyzed in any strain or natural population of this species so far. In this study, we obtained transcriptomic information which allowed us to characterize important gene families despite the absence of an annotated genome. Furthermore, we performed a quantitative analysis of gene expression after deltamethrin intoxication in a low resistant population. The results provided here increase the knowledge on detoxification processes in vectors of Chagas disease, which is essential for the design of new vector control strategies.

## Introduction

Chagas is a neglected tropical disease that can provoke disability and death. Six million inhabitants of Latin America are infected, whereas a fifth of the population in this region remains at risk. In the last decades, human migrations spread Chagas all over the world (https://www.paho.org/es/temas/enfermedad-chagas). The causative agent of the disease is the protozoan *Trypanosoma cruzi*, primarily transmitted to humans by feces of triatomine bugs deposited during blood-feeding. Vectorial transmission of *T. cruzi* occurs in wide zones of the Southern Cone, where *Triatoma infestans* is the primary vector [1]. In spite of its medical relevance, the annotated genomic sequence of *T. infestans* has not been published to date. The transcriptomic data available was obtained from particular body structures [2–4], or from a normalized library [5].

Given the absence of vaccines and efficient treatments during the chronic stage of Chagas disease, the control of triatomine populations is crucial for reducing vectorial transmission. Pyrethroid insecticides have been used for the control of triatomines for more than 20 years, given their efficiency and favorable toxicological properties [6]. Even though other types of compounds were experimentally tested [7], commercial formulations to replace pyrethroids for the control of triatomines are not available to date. In this context, the existence of pyrethroid resistant populations of *T. infestans* in regions from Argentina and Bolivia represents a public health problem [6]. The higher resistance ratio levels described in this species are associated with mutations in the insecticide target site, the voltage-gated sodium channel [8,9]. Other resistance-associated mechanisms are reduced penetration of insecticide due to cuticular alterations [10] and enhanced detoxification metabolism [11–16]. A focus located in Güemes Department (Chaco Province, Argentina) in the Gran Chaco ecoregion is of particular relevance given its extent and high levels of insecticide resistance detected in some triatomine populations [17]. In this region, vectorial transmission of Chagas could not be stopped, and a mosaic of *T. infestans* populations presenting either susceptible, low or high pyrethroid

resistant profiles co-exist, probably due to a non-homogenous use of pyrethroids in the area [17].

The quantitative study of the modulation of gene transcription in response to an insecticide could give cues for understanding both detoxificant response and insecticide resistance/susceptibility. In this sense, high-throughput RNA-Seq represents an approach to identify expression changes that could be involved in detoxification mechanisms, e.g. comparing transcription profiles of susceptible and resistant populations (as for example in [18–20]). Recently, the analysis of transcriptomic databases allowed the identification of new relevant components of the detoxifying response to insecticides in mosquitoes [21–23]. Studies addressing the transcriptomic response after a treatment with an insecticide in hemipterans are scarce (see for example [24,25]) and none of them have been performed in a vector of Chagas disease to date.

Studies on insect detoxification have been mainly focused on three protein superfamilies: cytochromes P450 (CYPs), carboxyl-cholinesterases (CCEs) and glutathione transferases (GSTs). These families have been previously characterized in triatomine species [14,26]. However, other protein families involved in stressful and/or toxic stimuli responses were not comprehensively analyzed in vectors of Chagas disease. These families include chemosensory proteins (CSPs), which have been studied in triatomine insects [27–29] but not in the context of detoxification, heat-shock Proteins (HSPs) and ATP-binding cassette (ABC) transporters. Research on the role of these proteins in the response to xenobiotics can provide interesting information on susceptibility and resistance to insecticides in triatomines.

Chemosensory proteins are small soluble proteins present only in arthropods [30]. Until very recently, the assigned role of insect CSPs was reduced to olfaction through the transport of hydrophobic odorant molecules, but recent evidence points to other functions (reviewed in [31]). Among them, a possible role in insecticide binding was suggested for these proteins [22,32], and growing evidence points to a CSP role in the response to xenobiotics: an overexpression of CSPs has been observed in several insect orders after a toxic stimulus [23,33–35] and in a pyrethroid-resistant mosquito population [22]. Moreover, a role of these proteins in insecticide susceptibility [36,37] and resistance [22] was recently proposed. All this recent evidence strongly indicates that, in addition to their involvement in the transport of odorant molecules, a role in insecticide susceptibility and/or resistance should be considered for CSPs.

Heat-shock proteins, also known as molecular chaperones, are involved in essential processes in the cell such as protein homeostasis. Their expression can be constitutive or induced by a wide variety of stressors, and are classified according to their sequence homology, molecular weight and function (reviewed in [38]). Among them, HSP70s can be classified as canonical and atypical, having the latter activity as co-chaperones [39]. A modulation in the expression of HSP70s in response to different toxic stimuli has been observed in insects [23,40–42] and an upregulation of HSP70 expression was observed in a pyrethroid resistant mosquito population [43]. A role of these proteins in resistance to starvation was proposed in *T. infestans* [44]. A similar observation was made in *R. prolixus*, where a role of these proteins in survival under starvation but also after feeding was proposed, and the expression of HSP70 transcripts was induced after thermal stress and feeding [45].

ABC transporters are membrane proteins which mediate the movement of substrates using ATP. Two types of ABC transporters have been described: 1) Full transporters (FT) are functional and possess two nucleotide binding domains (NBDs) and two transmembrane domains (TMDs); 2) Half transporters (HT) that contain one NBD and one TMD domain, and require homo or heterodimerization to be functional (reviewed in [46]). This superfamily is further classified into eight subfamilies (A-H) based on the homology of their NBDs [47]. A new subfamily (ABCJ) was recently proposed in *Ae. aegypti* [48]. Despite their nomenclature, some

subfamilies such as ABCE and ABCF lack transporting roles as they only encode for NBD domains. ABC superfamily, which is much less studied in arthropods than in other organisms such as bacteria and vertebrates, has been associated to a wide variety of physiological functions including the excretion of toxic compounds (reviewed in [49]). In insects, these transporters have been linked to insecticide transport and/or resistance, especially those belonging to subfamilies B, C and G [50].

Target-site mutations seem to be the main resistance mechanism in *T. infestans* high resistant populations (resistance ratio, RR, >100; [8,9,51]). For this reason, we hypothesized that modulation of gene expression could be less relevant for these populations, compared to the low resistant ones (RR = 2 to 10). Hence, in the present study we used RNA-Seq to analyze the transcriptomic response of a low-pyrethroid resistant population of *T. infestans* after topical application with the pyrethroid insecticide deltamethrin.

Our approach allowed us to find, through the evidence of their differential expression, possible candidates to be involved in the detoxification response of this insect, related to a wide variety of processes such as metabolism, cuticular rearrangements, transport and transcription/translation-related processes. We also performed a comprehensive characterization of some gene superfamilies that could be involved in this response, such as CSPs, HSPs and ABC transporters. The obtained results extend the genetic knowledge about *T. infestans* and provide information of the initial transcriptomic response to a pyrethroid in a low-resistant population. To our knowledge, this is the first RNA-Seq experiment to study the detoxificant response in a Chagas disease vector, and the first quantitative transcriptomic study in *T. infestans*.

## Methods

### Ethics statement

Pigeons used in this study were housed, cared, fed and handled in accordance with resolution 1047/2005 (Consejo Nacional de Investigaciones Científicas y Técnicas, CONICET, Argentina) regarding the national reference ethical framework for biomedical research with laboratory, farm, and nature collected animals (Marco Ético de Referencia para las Investigaciones Biomédicas en Animales de Laboratorio, de Granja y Obtenidos de la Naturaleza), which is in accordance with the standard procedures of the Office for Laboratory Animal Welfare, Department of Health and Human Services, NIH and the recommendations established by the 2010/63/EU Directive of the European Parliament, related to the protection of animals used for scientific purposes and National Law 14,346 on Animal Welfare. The final protocol was evaluated by a committee in CREG, which confirmed the accordance with the ethical frameworks. Biosecurity considerations are in agreement with CONICET resolution 1619/2008, which is in accordance with the WHO Biosecurity Handbook (ISBN 92 4 354 6503). The collection of insects in dwellings was performed in agreement with the Argentinean National Health Ministry ethic requirements.

### Insects

Insects were a third generation descendant (F3) from a field population collected in Colonia Castelli (CC, 26°0′38′′S, 60°39′18′′W), a small village near to Juan José Castelli, the main city of Güemes Department, in the province of Chaco, Argentina. The first generation descendant of this field population was previously classified as low resistant to deltamethrin, according to the definition adopted by Fronza *et al*. [17]. The LD50 (the dose that kills 50% of the first instar insects after 24 h) of this population was 0.56 ng/ insect [17] and the LD30 was 0.2 ng/ insect. Resistance ratio ($LD_{50}$ of the population/ $LD_{50}$ of the reference susceptible colony) determined

for this population was 3.06 [17]. Colonia Castelli population was selected for this study based on its geographical origin: an area in which high levels of resistance to pyrethroids have been detected in this species. The *T. infestans* colony was raised at the laboratory under controlled temperature (26 ± 1C), humidity (50–70%), and a photoperiod of 12:12 (L:D) h. A pigeon was weekly provided as a blood meal source [52].

## Sublethal intoxication assay and sample preparation

Starved first instar nymphs 5–7 days after egg hatching were topically treated on the dorsal abdomen with 0.2 ul of 0.001 mg/ml of deltamethrin (Sigma-Aldrich, St Louis, MO, U.S.A.) in acetone (Merck, Buenos Aires, Argentina) (LD30), or pure acetone for controls. A 10 μl Hamilton syringe provided with a repeating dispenser was used. After four hours, insects were immersed in Trizol reagent (Thermo Fisher Scientific, USA) and conserved at -80˚C until RNA extraction. Each experimental replica consisted in a pool containing 12 to 15 insects. Four replicates were obtained for each experimental group (deltamethrin or acetone treated). Each sample was homogenized using plastic pestles. Total RNA was extracted using Trizol reagent (Thermo Fisher Scientific, USA) according to the manufacturer's instructions.

## Sequencing and data preprocessing

Library construction and high-throughput sequencing services were hired at Novogene Corporation Inc. (Sacramento, USA). A total of eight cDNA libraries (four *per* experimental condition) were constructed using the NEBNext Ultra RNA Library Prep Kit (New England Biolabs, USA). The libraries were sequenced using Illumina HiSeq 2500 (paired-end reads with 150 bp length). The raw sequence data set is available at the National Center for Biotechnology Information (NCBI) with the SRA Bioproject number PRJNA778244. To evaluate the quality of the resulting reads and the presence of sequencing adapters, FASTQC tool (available at www.bioinformatics.babraham.ac.uk/projects/fastqc) was used. The software Trimmomatic v0.36 [53] was used in the paired-end mode to remove low quality bases from 5' and 3' ends with the parameters TRAILING: 5 and LEADING: 5. Besides, the SLIDING-WINDOW parameter was set as 4:18 and the adapters were removed using the parameters ILLUMINA-CLIP:TruSeq3-PE-2.fa:2:30:10. Only reads longer than 50 bp were maintained.

## Transcriptome *de novo* assembly, prediction of coding regions and quality assessment

The resulting trimmed and quality-filtered paired-end reads of the eight samples were used to construct a *de novo* transcriptome assembly using the software Trinity v2.10.0 [54,55] with default parameters. The basic statistics of the resulting assembled transcriptome were assessed with the *TrinityStats.pl* script.

*De novo* transcriptome assemblies contain thousands of assembled transcripts, putative isoforms and chimeric transcripts that can affect subsequent analyses, like transcript identification and differential expression analysis. For this reason, and in order to reduce the redundancy and complexity of our database, we decided to extract the predicted coding sequences (CDS) and to cluster them according to their sequence identity. Coding regions of the assembled transcripts were predicted using TransDecoder software v5.5.0 (http://transdecoder.github.io). Open reading frames (ORFs) with a product of at least 100 amino acids were predicted using the *TransDecoder.LongOrfs* script. BLASTp (v2.5.0+) and HMMscan (v3.2.1) searches were performed on the predicted ORF database using the complete UniProtKB/Swiss-Prot and Pfam-A databases as queries, with an e-value cutoff of $1e^{-50}$. The results of these searches were used in the *TransDecoder.Predict* script to obtain the

predicted CDS. To avoid redundant CDS, the resulting sequences were then analyzed with the software CD-HIT v4.8.1 in cd-hit-est mode [56] to cluster them considering a sequence identity threshold of 0.95. This non-redundant CDS database was used to perform the following analyses. Finally, to assess the completeness of this database, BUSCO v4.1.4 [57] was used in protein mode against the hemiptera_odb10 lineage dataset.

## Analysis of voltage-gated sodium channel transcripts

To identify the transcripts encoding the fragment of the voltage-gated sodium channel (the target site of pyrethroids) where pyrethroid-resistance associated mutations have been described [8,9], a BLASTp search was performed in the translated non-redundant CDS dataset. The protein fragments of this sequence previously identified in *T. infestans* [9] were used as queries, with an e-value cut-off set at $1e^{-5}$. To assess the presence of mutations in the fragment of interest on the identified sequences, the trimmed paired-end reads from each sample were mapped to the non-redundant CDS dataset using STAR software v2.7.1a [58] (with *outFilterMultimapNmax* set at 50, and *outFilterScoreMinOverLread* and *outFilterMatchNminOverLread* set at 0.3). The resulting bam files (sorted by coordinate) were merged and indexed using Samtools software v1.10, and visualized along with the reference using the software IGV v2.9.2. Finally, a nucleotide alignment was constructed using Clustal Omega [59] to compare the identified fragment along with previously reported sequences.

## Transcript quantification and differential expression analysis

The non-redundant CDS database was used as a reference to quantify transcript abundance in every sample with Salmon software v1.4.0 [60], using the *align_and_estimate_abundance.pl* Trinity script and the trimmed paired-end reads from each sample. The matrix of counts *per* transcript in each sample was reported with the *abundance_estimates_to_matrix.pl* Trinity script. The obtained matrix was filtered using the HTSFilter v.1.28 [61] in RStudio to eliminate transcripts with low expression and/or high variation. Following, the filtered matrix was used to perform differential expression analysis using DESeq2 v.1.28 [62] with the *lfcShrink* function. This is based on the "Approximate Posterior Estimation for Generalized Linear Model" method that utilizes an adaptive Bayesian shrinkage estimator to obtain more accurate log2 fold-change values [63,64]. Differentially expressed (DE) genes were identified based on a False Discovery Rate (FDR) < 0.05 and submitted to BLASTx searches against: 1) UniProtKB/ Swiss-Prot, 2) the predicted protein sequences from *D. melanogaster* (release FB2021_02, downloaded from FlyBase (flybase.org)); 3) the predicted proteins from *R. prolixus* (release 51, downloaded from VectorBase [27,65]). Additionally, HMMscan searches on protein sequences were performed using the Pfam-A database. The output of these searches was integrated using Trinotate v3.2.1 (https://trinotate.github.io), with an e-value of $1e^{-5}$ to extract the positive hits from BLAST searches, and using domain noise cutoff as threshold to report PFAM results. Additionally, BLAST searches against the non-redundant (nr) database from NCBI were made, identifying four sequences in the DE set with hits on non-insect sequences that were discarded from the analysis.

A heatmap of the DE set was created with pheatmap v.1.0 in RStudio, with count data transformed using the *rlogTransformation* function included in DESeq2 [62]. A dendrogram was plotted with hierarchical clustering among genes based on Euclidean distance and ward.D2 method for clustering.

A summary of the main steps of the described pipeline used for data curation and analysis is available in S1 File.

## Characterization of gene families and sequence analysis

HMMscan searches (with an e-value threshold for reporting models set at 1e-5) were performed on both the translated non-redundant CDS database from *T. infestans* and in the annotated proteins from *R. prolixus*, available at VectorBase (version 51; [27,65]). Additional BLAST searches were conducted with an e-value cutoff set at 0.01, using tBLASTn to analyze the non-redundant CDS database from *T. infestans* and BLASTp to analyze the *R. prolixus* predicted proteins. The following PFAM seed alignments (used as queries for BLAST) and PFAM profile HMMs (used for hmmscan searches) were used for each family: PF00005 for ABC transporters, PF00012 for HSP70 proteins and PF03392 for CSPs (searched only in *T. infestans*). Iterative searches were conducted using resulting sequences as queries until no new sequences were identified. Finally, BLAST searches against the non-redundant database from NCBI were made and transcripts with hits on non-insect sequences were discarded from the analysis.

Partial transcriptomic sequences were manually analyzed and corrected if evident assembly and/or sequencing errors were detected, using sequences from other insects for comparisons. These manually curated sequences were only used for the sequence and phylogenetic analyses. To avoid redundancy, CD-HIT v4.8.1 was used to analyze partial sequences and only those with less than 90% of amino acid identity with other sequences in the family were kept. The CSP sequences previously reported for *T. infestans* [5] were manually curated as described and clustered with the set found in this study using CD-HIT (in cd-hit-est mode, using a sequence identity threshold of 0.95) to obtain a non-redundant set for further analysis. In the case of ABC transporters, sequences from A to H subfamilies were analyzed.

The final sequence set was analyzed with the following softwares: TOPCONS2 [66] for the prediction of transmembrane helices in the transporter proteins; Interproscan v85.0 [67] to analyze protein domains, and SignalP v5.0 [68] to assess the presence of signal peptide. In the case of CSPs, Clustal Omega [59] was used to align *Drosophila melanogaster* CSP sequences along with the predicted proteins to assess the presence of conserved residues and PSIPRED v4.0 [69] was used to predict CSPs secondary structure. Transcripts Per Million (TPM) values were used in gplot package (v.3.1.1) in RStudio to construct expression heatmaps for the gene families of interest. A dendrogram was plotted with hierarchical clustering among genes based on Euclidean distance and the complete linkage method for clustering.

## Phylogenetic analysis

The protein sequences identified in *T. infestans* and *R. prolixus* were aligned with MAFFT [70] along with sequences previously reported in *D. melanogaster* for ABC transporters [47] and HSP70 (FlyBase gene groups FBgg0000497 and FBgg0000498). In the case of the CSP family, sequences previously reported for *R. prolixus* [27,29] and *T. brasiliensis* [28] were aligned along with *T. infestans* predicted proteins. For this, G-INS-i strategy with the following settings was used: unaligned level = 0.1; offset value = 0.12; maxiterate = 1000 and the option *leave gappy regions*. The resulting alignments were trimmed using trimAl v1.2 [71] with default parameters except for the gap threshold, which was fixed at 0.3. Following trimming, IQ-tree v1.6.12 [72] was used to build phylogenetic trees based on the maximum-likelihood approach. The branch support was estimated using the approximate Likelihood Ratio Test based on the Shimodaira-Hasegawa-like procedure (SH-aLRT, [73]). The best-fit amino acid substitution models, selected by ModelFinder [74] within IQ-tree and chosen according to the Bayesian Information Criterion, were LG+I+G4 for CSPs, LG+G4 for HSPs and LG+F+R7 for ABCs. The phylogenetic trees were visualized and edited with iTol online tool [75] and rooted at midpoint.

## Results and discussion

### RNA-Sequencing, *de novo* assembly and prediction of coding regions

The sequencing output of the eight samples generated more than 185 million paired-end reads of 150 bp length of raw data (Table A in S2 File). The resulting *de novo* assembled transcriptome contained a total of 431,511 transcripts and 292,177,510 assembled bases, with a GC percent of 34.18.

A total of 63,890 predicted CDS were identified in the *de novo* assembled transcriptome and 33,321 of them were kept after removing redundancy. The BUSCO searches revealed more than 96% of complete BUSCOs (S1 Fig), showing the completeness of our database.

### Analysis of target-site mutations associated to pyrethroid resistance

Although *T. infestans* population from CC used in this work was classified as low-pyrethroid resistant, it is surrounded by localities where both susceptible and resistant populations were detected [17]. It is known that high pyrethroid resistance in *T. infestans* is associated with the fixation of target site (*kdr*) mutations in a population [8,9]. Hence, we hypothesized that the low resistance levels reported in CC population could be explained by the presence of a *kdr* mutation with middle or low frequency, as a consequence of insect migrations from neighboring localities. To address this hypothesis, we identified all the reads that mapped on transcripts encoding a fragment of the domain II of the voltage-gated sodium channel (S2 Fig), which is the target site of pyrethroids. Two point mutations associated with pyrethroid resistance (named L925I and L1014F) are located in this region of the gene [8,9]. All the detected reads have the wild-type (susceptible) sequence, indicating that the low resistance levels observed in CC are not explained by target site mutations. A similar result was found in Fronza *et al.* (2020) [16], where both substitutions were absent in CC insects analyzed. Interestingly, a silent substitution from thymidine to cytosine was detected in 63% of the mapped reads (base pair 183 in S2 Fig). This substitution was also found in high proportion in a resistant population from a neighboring locality, which carries the non-silent *kdr* mutation L925I, and in low proportion in the susceptible population [9]. In addition, a microsatellite study performed in the zone proposed some degree of gene flow between CC and a near high-resistant locality [76]. This reinforces the hypothesis that the ancestral *T. infestans* populations from the region had a common genetic background. The emergence of populations with different levels and mechanisms of resistance could have been promoted by discontinuities in the insecticide spraying and enhanced by the effect of environmental factors [77].

### Differential expression analysis

Given the absence of target-site mutations, other detoxifying responses could be responsible for the low level of resistance observed in the population under study. The quantitative analysis of the transcriptomic response to a low dose of deltamethrin could help to identify genes and gene families potentially involved in detoxification processes; however, functional validations will be necessary to confirm the role of these differentially expressed genes.

Among all eight sequenced samples, more than 70 million reads mapped back to the coding regions of the assembled transcriptome (Table B in S2 File). The differential expression analysis showed that 77 genes significantly changed their expression levels after deltamethrin treatment (FDR < 0.05); 61 of them were upregulated and 16 were downregulated (Figs 1 and S3, Table C in S2 File). More than 10 DE genes encode for uncharacterized proteins; most of them show high similarity to *R. prolixus* sequences (Table D in S2 File), indicating that they are putative proteins and not artifacts of the bioinformatic predictions.

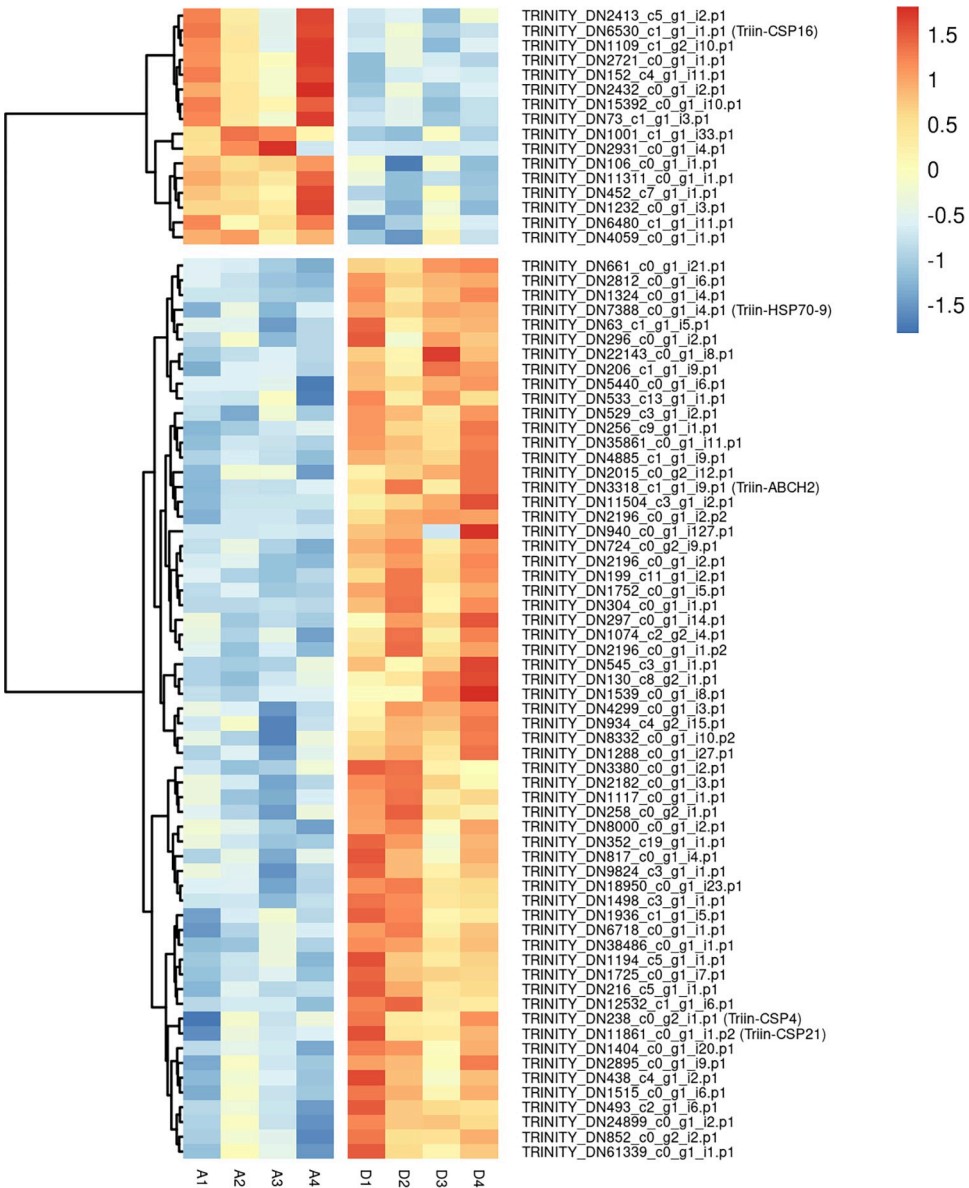

**Fig 1. Effect of deltamethrin treatment on the transcription of differentially expressed genes identified in *T. infestans*.** Heatmap was plotted using z-score calculated from the transformed count data, by means of a color scale in which blue/red represent lowest/highest expression. The dendrogram on the left represents the results of the row hierarchical clustering. A: Acetone (control) samples. D: Deltamethrin-treated samples.

Several cellular processes were affected after the insecticide application: among others, DE genes related to transcription and translation processes, rearrangement of the cuticle and transmembrane transport were identified (Table D in S2 File).

**Transcription and translation processes.** More than 20% of the DE genes encode proteins related with transcription and translation processes and their regulation (Table D in S2 File). From them, three upregulated genes possess a basic leucine zipper (bZIP_1) domain (Table D in S2 File). Proteins belonging to bZIP family have been related to detoxification and insecticide resistance in other insects [78–83]. One of the transcripts that contains this domain is TRINITY_DN1498_c3_g1_i1.p1, which seems to be an homologue of the *Cyclic-AMP*

*response element binding protein B* (CREB) (Fig 1 and Table D in S2 File). Interestingly, the over-expression of this transcription factor was also observed in an imidacloprid resistant strain of *Bemisia tabaci*, and it was proposed that its phosphorylation has a key role in the resistance to this insecticide through the upregulation of a cytochrome P450 [81]. Another upregulated gene (TRINITY_DN1194_c5_g1_i1.p1), which encodes an helix-loop-helix DNA-binding domain (PF00010) (Fig 1 and Table D in S2 File), seems to be related to the *D. melanogaster* transcription factor *diminutive*. This gene was previously implicated in pyrethroid resistance in *Anopheles gambiae* [21] and in the regulation of transcripts related to the response to insecticides [83]. Whether *CREB* and *diminutive* genes have a role in insecticide detoxification and resistance in *T. infestans* deserves to be studied.

As observed in other insect species (reviewed in [84]), a diverse set of transcription factors and other regulatory elements of *T. infestans* may induce an early transcriptomic activation that could further regulate genes directly involved in the global detoxifcant response. Nevertheless, functional studies and Chromatin Immunoprecipitation sequencing (ChIP-Seq) experiments would be necessary to confirm our results.

**Cuticule-related transcripts.**   Two upregulated transcripts are related to chitin metabolism (TRINITY_DN38486_c0_g1_i1.p1 and TRINITY_DN18950_c0_g1_i23.p1) (Fig 1 and Table D in S2 File). An upregulation of chitinase-encoding genes was previously observed in *Sogatella furcifera* after treatment with different types of insecticides, including deltamethrin [24]. The transcriptional changes seen in our study indicate that the rearrangement of the cuticle could be relevant in the response of *T. infestans* to the pyrethroid. This is in agreement with a role of the integument in detoxification and resistance to insecticides in *T. infestans* [10,15].

**Transmembrane transport.**   The abundance of six transcripts related to transmembrane transport was affected by the insecticide administration (Table D in S2 File). One of the upregulated transcripts belongs to ABC transporters family, which will be further analyzed here (TRINITY_DN3318_c1_g1_i9.p1) (Fig 1 and Table D in S2 File). Two of them (TRINITY_DN1232_c0_g1_i3.p1 and TRINITY_DN2432_c0_g1_i2.p1), whose expression decreases after treatment, are solute carriers that belong to the Major Facilitator Superfamily (MFS) (Fig 1 and Table D in S2 File). The underexpression of some of these transporters was also observed in *Ae. aegypti* larvae after intoxication with an essential oil [23]. These results, along with previous evidence [85,86], support the hypothesis that MFS could participate in the response to xenobiotics.

**Energetic metabolism.**   A significant difference in the abundance of some transcripts potentially related to lipid and carbohydrate metabolism was found in this study (Fig 1 and Table D in S2 File). Consistently, transcriptional changes related to lipid metabolism have been reported in *Ae. aegypti* larvae in response to toxics [23,87] and alterations in carbohydrate metabolism after pyrethroid treatment has been observed [88]. The regulation of energetic metabolism after intoxication in triatomines deserves to be studied.

In addition to these processes, a significantly higher abundance of a transcript (TRINITY_DN22143_c0_g1_i8.p1) that encodes for an odorant binding protein (OBP) was found after treatment with deltamethrin (Fig 1 and Table D in S2 File). In agreement with our results, the overexpression of some OBPs was observed after treatment with dissimilar toxic compounds in insects from different orders (see for example [23,34,87]). Moreover, the role of OBPs in the defense against xenobiotics was demonstrated in *Tribolium castaneum* [36,89]. Three members of *T. infestans* CSP family were also found DE (see details below). Our results reinforce the idea of the involvement of some OBPs and CSPs in the response to xenobiotics. Finally, the overexpression of one HSP belonging to the HSP70 family was observed (Fig 1 and Tables C and D in S2 File); a complete analysis of these proteins is presented below.

## Characterization of selected detoxification-related gene families

The analysis of transcription modulation after intoxication carried out in this study did not reveal a differential expression of CYP, GST or CCE genes, although the expression of some members of these families was reported to be induced after insecticide treatment (see for example [24,90–92]). Quantitative PCR showed an induction of CYP genes in both fat body [13] and integument [15] in *T. infestans* after intoxication with deltamethrin. The fact that a differential expression of these genes is not observed in this study may be due to differences in the populations, doses and tissues analyzed. Previous transcriptomic studies on *Ae. aegypti* larvae did not detect changes in the expression of GST, CYP or CCE exposed for 48 h to sublethal doses of the pyrethroid permethrin [87]. However, a time dependent modulation of transcripts belonging to these gene families was observed for *Anopheles stephensi* larvae at different times post-exposition to permethrin [93] and for a resistant population of *Anopheles coluzzii* after sublethal exposure to deltamethrin [94]. Moreover, genes belonging to the CYP superfamily were found overexpressed in pyrethroid resistant populations, including some of *T. infestans* [13–15]. The results obtained in this work, in agreement with previous evidence, suggest that the genetic modulation of detoxifying enzymes triggered by pyrethroids may depend on the dose, the exposure time and the species and populations analyzed.

The hypothesis of an involvement of CSPs, HSPs and ABC transporters in the response to insecticides and resistance has received an increasing amount of scientific evidence in the last years [21,23,50]. The analysis performed here also suggests their relevance in the early response to deltamethrin in *T. infestans*. Hence, we characterized these gene families in the assembled transcriptome, focusing on their expression modulation. Furthermore, we analyzed their phylogenetic relationships with sequences from other insect species, including *R. prolixus*. Although this assembled transcriptome shows a high degree of completeness, the availability of *T. infestans* genome annotations will be necessary to confirm the number of genes belonging to each of these families, as well as to assess if the reported sequences belong to different genes or represent allelic variants or isoforms product of alternative splicing.

**Chemosensory proteins.**   The analysis of the predicted CDS revealed 18 complete CSPs in *T. infestans*, which encode proteins between 113 and 136 amino acid residues in length (Table E in S2 File). The characteristic features of these proteins (such as signal peptide, four conserved cysteine residues and six α-helices, reviewed in [95]) were identified in most of these sequences, confirming the identity and completeness of the assembled CSP sequences. In addition, four partial sequences were detected; one of them encoded a very short protein (54 amino acids) that was excluded from the phylogenetic analysis (Table E in S2 File). Hence, considering partial and complete sequences, the complement of CSPs in *T. infestans* has at least 22 transcripts. This would give for *T. infestans* a greater number of CSPs than those found in *R. prolixus* genome (19) [27] and in a *T. brasiliensis* transcriptome (16) [28]. Interestingly, the size of the CSP complement varies across different insect orders [30], with variations observed within insect families, such as the case of Culicidae where it goes from 4 (*Anopheles darlingi*) to 83 (*Aedes albopictus*) [96]. The results obtained here along with previous evidence indicate a lower variation in CSP number in triatomines.

The phylogenetic analysis revealed that CSP family is conserved in the subfamily Triatominae: with the exception of Triin-CSP20 and Triin-CSP21, all *T. infestans* CSPs have an orthologue in *R. prolixus* and/or *T. brasiliensis*. This suggests an evolution pressure to conserve both the structure of the family and the sequence of CSPs across the kissing-bug evolution (Fig 2). Whereas 11 CSP members presented 1:1:1 orthologies in *T. infestans*, *T. brasiliensis* and *R. prolixus*, sequences such as CSP14-CSP15 from *T. infestans* and *T. brasiliensis* grouped with orthologous sequences from *R. prolixus* that might indicate independent duplications in *R.*

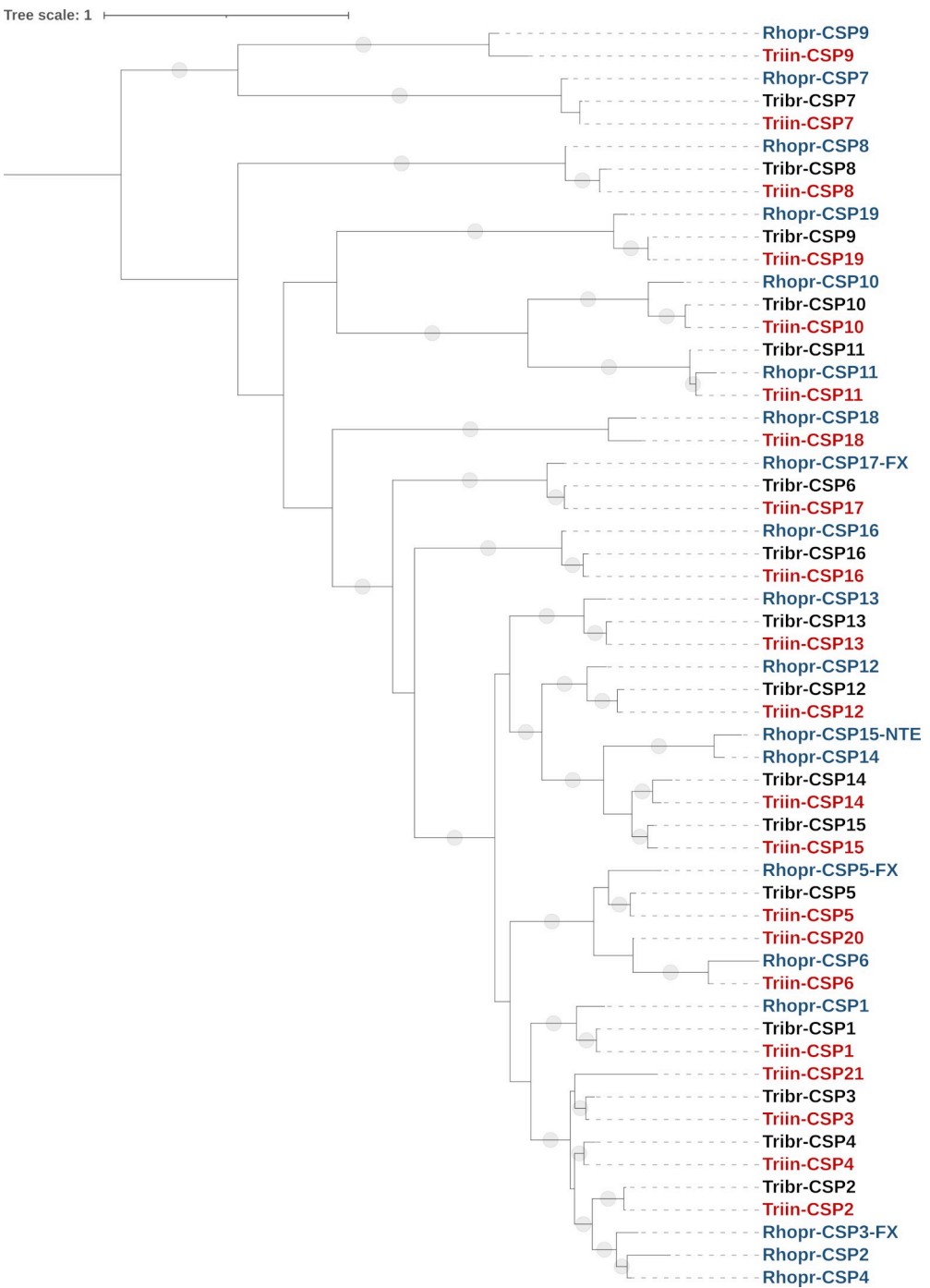

**Fig 2. Phylogenetic tree of the CSP family from *T. infestans*, *T. brasiliensis* and *R. prolixus*.** The maximum-likelihood tree was constructed using IQ-Tree and protein sequences obtained from the transcriptomes of *T. infestans* (Triin-, red, available in Table E in S2 File) and *T. brasiliensis* (Tribr-, black, obtained from [28], and from the *R. prolixus* genome (Rhopr-, blue, NTE: N-terminus missing in gap; FX: gene model repaired based on *de novo* transcriptome assemblies; obtained from [29]). Sequences from *T. infestans* were named according to their *R. prolixus* orthologues, with the exception of Triin-CSP20 and Triin-CSP21 which have no orthologues in *R. prolixus*. Branches with SH-aLRT support values > 80 are marked with a grey dot. The tree was rooted at midpoint.

*prolixus* and the *Triatoma* species. The CSP2 sequences from *T. infestans* and *T. brasiliensis* show a close phylogenetic relationship with the paralogues CSP2-4 from *R. prolixus*, whereas the sequences CSP3 and CSP4 from *T. infestans* and *T. brasiliensis* seem to lack homologues in *R. prolixus*. The partial sequence Triin-CSP21 has no clear orthologies in other species and lacks 2 conserved cysteine residues (Table E in S2 File). Genomic information will be necessary to confirm if these sequences come from different genes in *Triatoma* species and the analysis of CSP family including other triatomine species would be useful to elucidate if these gene duplications are extended across *Triatoma* and *Rhodnius* genus.

Triatoma infestans* CSP4 and CSP21 were found upregulated, whereas CSP16 was downregulated in response to deltamethrin (Fig 1), pointing to a differential modulation of expression within the CSP family. Even though previous studies showed that CSPs tend to be upregulated during the detoxificant response (see for example [22,23]), a member of the CSP family was also downregulated in response to imidacloprid in *Ae. aegypti* [87], and a number of CSPs were downregulated at different time points after a treatment with deltamethrin in *An. gambiae* [22]. Given that different members of the CSP family show variations in their ability to bind a particular xenobiotic [22,23], an adaptive response to intoxication could be proposed. In this way, the global expression of CSPs could be shifted after intoxication, in order to prioritize the expression of those members with higher ability to bind a particular toxic molecule. Interestingly, the *T. brasiliensis* orthologues of two DE CSPs (Tribr-CSP16 and Tribr-CSP4) were found underexpressed in domiciliary bugs in comparison to peridomiciliary individuals [28], supporting a role of these CSPs in the adaptation to environmental changes.

Chemosensory proteins were clustered in 5 groups based on their expression patterns (Fig 3). The group containing the upregulated sequences CSP4 and CSP21 (along with CSP5 and CSP2) show the highest expression in both treated and control groups. Interestingly, CSP2, CSP4 and CSP21 are phylogenetically related (Fig 2). The high conservation at the sequence level and a similar expression pattern could indicate that genes encoding these CSPs are organized in genomic clusters in *T. infestans*, as previously observed for CSPs in insect genomes [30]. The group integrated by CSP7, CSP12, CSP16 and CSP17 is the second most expressed, followed by the group integrated by CSP1, CSP8, CSP11 and CSP15 and the group integrated by CSP10, CSP13, CSP19 and CSP20. The group containing CSP6, CSP9 and CSP14 shows the lowest expression levels within the family in both analyzed conditions (Fig 3).

**Heat-shock proteins 70.** We found 10 complete ORFs identified as HSP70 on the *T. infestans* transcriptome, and one of them, named Triin_HSP70-9, was significantly upregulated (Tables C and D in S2 File). We also performed an analysis of the HSP70 family in *R. prolixus*, where they have not been comprehensively characterized so far despite the availability of its genome. In this species, 13 HSP70 candidates were detected; six of them with complete CDS, and seven partial sequences (including one chimeric protein, RPRC003073-PA), probably due to errors in the automatic gene prediction process (Table F in S2 File). The number of HSP70 found in *T. infestans* and *R. prolixus* is similar to the numbers found in mosquitoes, which ranged between 7 and 11 [43].

Most *T. infestans* HSP70 proteins possess the HSP70 domain, the nucleotide binding domain, the peptide-binding domain and the C-terminal domain, with the exception of Triin_HSP70-1 and Triin_HSP70-2 (Table F in S2 File). The presence of HSP70 domain was confirmed in all *R. prolixus* sequences, with the exception of the chimeric protein and the sequence RPRC002179-PA, which encodes a protein with less than 100 amino acids. The sequences RPRC011930-PA and Triin-HSP70-2 represent a complete ORF but lack the peptide-binding and the C-terminal domains (Table F in S2 File).

The phylogenetic analysis included *T. infestans* and *R. prolixus* sequences, with the exception of the *R. prolixus* chimeric sequence and those shorter than 250 amino acids. It shows 6

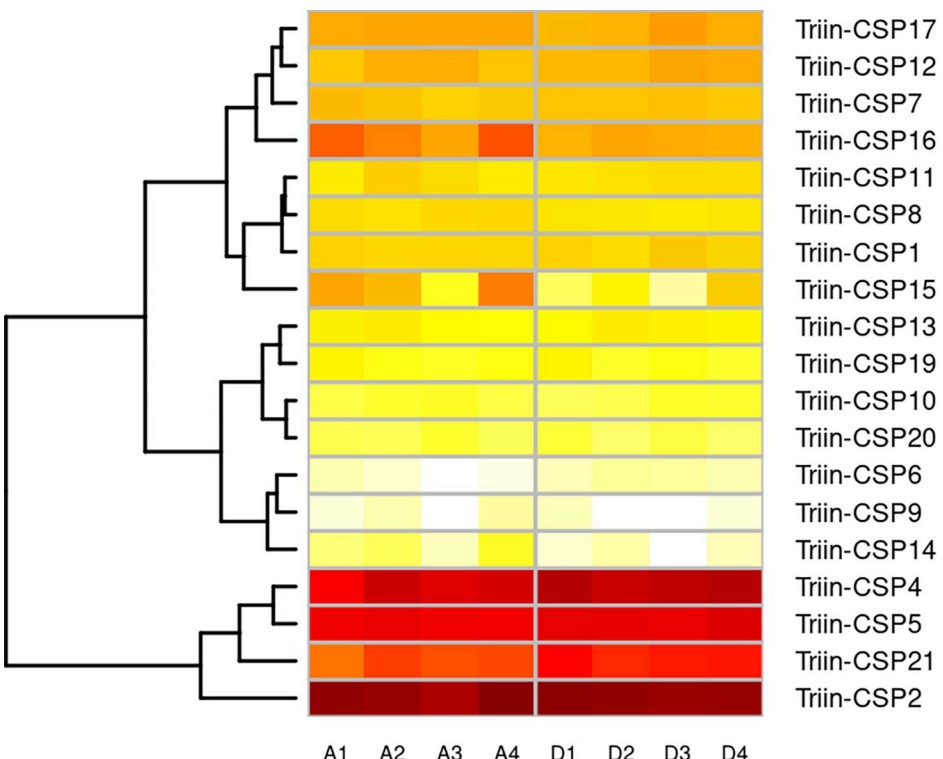

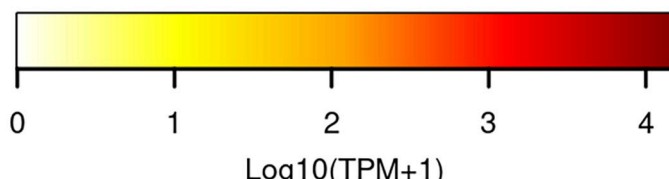

**Fig 3. Effect of deltamethrin on the transcription of *T. infestans* CSPs.** The heatmap was created using Transcript per Million (TPM) as input of the gplot R package. Transcript abundance was represented as $Log_{10}$(TPM +1) where white/red represents the lowest/highest expression. A dendrogram was plotted using a hierarchical clustering of gene expression values based on Euclidean distance and complete linkage method for clustering. A: Acetone (control) samples. D: Deltamethrin-treated samples. Expression data of *T. infestans* CSP3, CSP18 and CSP22 is not available because these sequences were obtained from [5] and they were not detected in the transcriptome generated in this work.

clades that represent 1:1 orthologies between *R. prolixus* and *T. infestans* sequences (Fig 4). Moreover, two sequences from *T. infestans* and two from *R. prolixus* do not show a clear orthology with any other sequence in the phylogenetic tree. The sequences Triin-HSP70-3 and Triin-HSP70-8 have orthologues in *D. melanogaster* but not in *R. prolixus*; the correction of the partial sequences in the predicted proteins of *R. prolixus* will allow us to assess the presence of these sequences in the species genome. The sequences Triin-HSP70-9 and Triin-HSP70-10 along with Rhopr-RPRC000930-PA and Rhopr-RPRC009759-PA are grouped together and related to a *D. melanogaster* clade which contains 7 HSP sequences; this could reflect specific expansions or contractions that deserve to be studied. The sequences Triin_HSP70-5 and

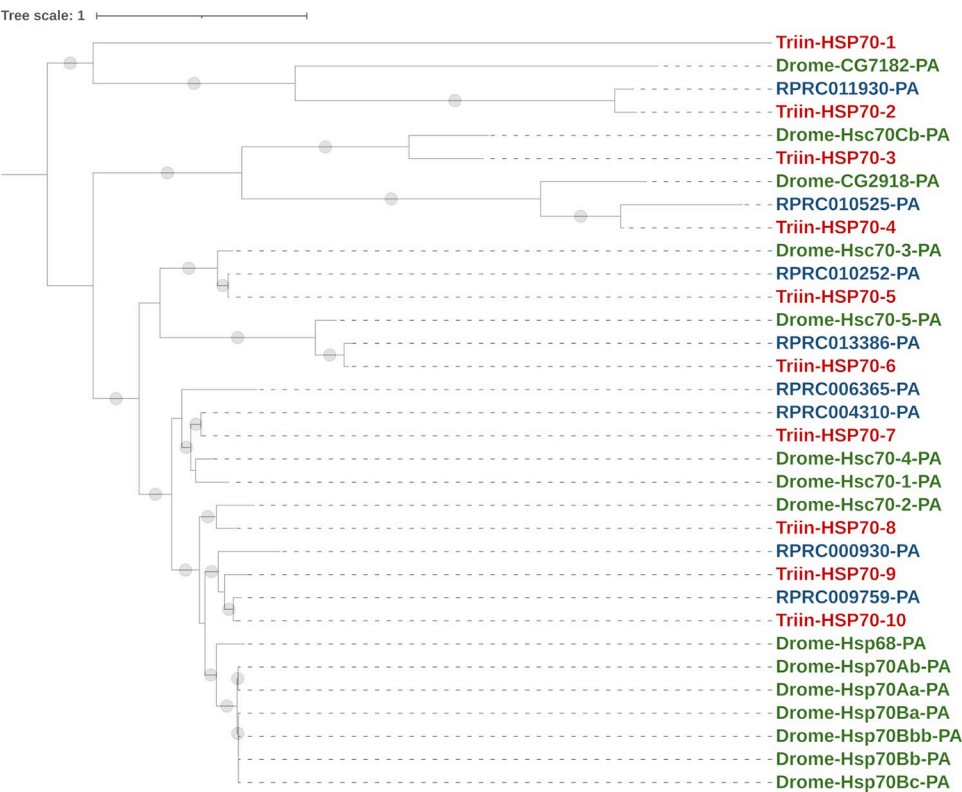

**Fig 4. Phylogenetic tree of the HSP70 superfamily from *T. infestans*, *R. prolixus* and *D. melanogaster*.** The maximum-likelihood tree was constructed using IQ-Tree and protein sequences obtained from *T. infestans* transcriptome (Triin-, red, available in Table F in S2 File) and from *R. prolixus* (RPRC-, blue, VectorBase ID is shown) and *D. melanogaster* (Drome-, green, FlyBase gene name and isoform are shown) genomes. *Rhodnius prolixus* chimeric sequence and those shorter than 250 amino acids were excluded from the analysis (Table F in S2 File). *Triatoma infestans* sequences were numbered according to their position in the tree. Branches with SH-aLRT support values > 80 are marked with a grey dot. The tree was rooted at midpoint.

RPRC010252-PA present domains related to BiP chaperone and are grouped with *D. melanogaster* BiP, Hsc70-3-PA. From the atypical HSP70 sequences from *D. melanogaster*, Dmel-CG2918-PA (orthologue of *HYOU* human gene), is grouped with RPRC010525-PA and Triin_HSP70-4, which possess PTHR45639:SF3 domain (Table F in S2 File), and Dmel-Hsc70Cb (HSP110) is grouped with Triin_HSP70-3 from *T. infestans* (Fig 4 and Table F in S2 File).

Expression analysis revealed that Triin_HSP70-5 and Triin_HSP70-7 show the highest expression within the family, followed by Triin_HSP70-6 (Fig 5). Triin_HSP70-2, Triin_HSP70-3 and Triin_HSP70-8 show an intermediate expression in both conditions, whereas Triin_HSP70-1, Triin_HSP70-4 and Triin_HSP70-10 show the lowest, along with Triin_HSP70-9. The latter is the only transcript that showed a significant induction 4 h after deltamethrin exposure; the analysis of additional times post-exposure and insecticide doses would be interesting to assess the expression pattern of the entire family in response to an intoxication.

**ABC transporters.** The ABC transporters superfamily has not been analyzed in detail in triatomine insects, with the exception of a recent analysis that reported the amount of ABC genes in *R. prolixus* [97]. In this work, we found 65 and 58 candidate sequences for ABC transporters in *T. infestans* and *R. prolixus*, respectively (Table G in S2 File), belonging to A-H subfamilies. The recently proposed ABCJ subfamily [48] was not analyzed here. A total of 21

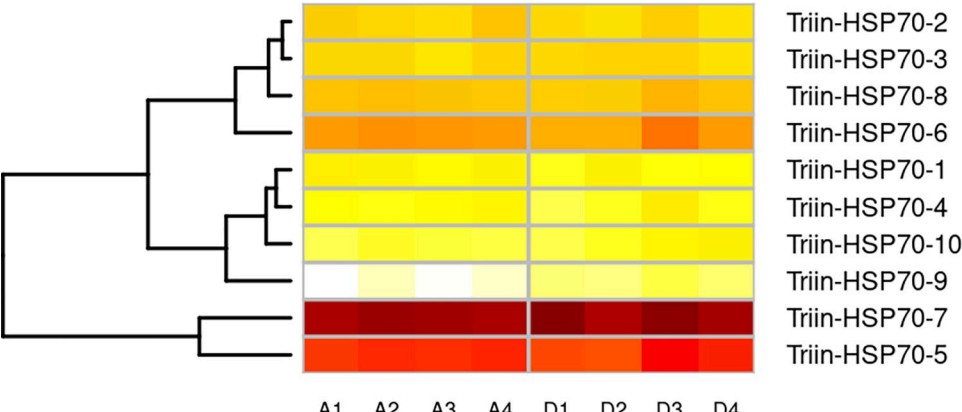

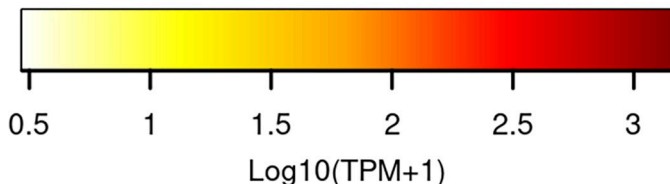

**Fig 5. Effect of deltamethrin on the transcription of the *T. infestans* HSP70 family.** The heatmap was created using Transcript per Million (TPM) as input of the gplot R package. Transcript abundance was represented as $Log_{10}(TPM+1)$ where white/red represents the lowest/highest expression. A dendrogram was plotted using a hierarchical clustering of gene expression values based on Euclidean distance and complete linkage method for clustering. A: acetone (control) samples. D: deltamethrin-treated samples.

candidates (10 from *T. infestans* and 11 from *R. prolixus*) were excluded from further analysis given that the PF00005 domain, characteristic of ABC transporters, was absent. According to our sequence and phylogenetic analyses, the ABC repertoire of *T. infestans* was composed of 55 sequences (45 complete and 10 partial) while *R. prolixus* presented 47 ABC transporters, being 10 complete and 37 partial/misannotated sequences (Table G in S2 File). A recent work identified a similar number of ABC genes in *R. prolixus* (49) [97]; this subtle difference between studies may be due to the different search strategies employed.

Subfamily G is the most abundant in both triatomines analyzed, with almost one third of the total ABC sequences (16 in *T. infestans* and 14 in *R. prolixus*, Fig 6 and Table G in S2 File). The expression of several members of ABCG subfamily was enriched in structures with predominance of cuticle in mosquitoes, such as legs in *An. gambiae* [98], and sensory appendages in *Ae. aegypti* [99,100]. This tissue enrichment is supportive of a role of ABCG in the transport of lipids to the cuticle, which has been confirmed by gene silencing in the beetle *Tribolium castaneum* [101].

Subfamilies C and H are abundant in *T. infestans* (8 ABCCs and 15 ABCHs), being ABCH more abundant than in *R. prolixus*, whereas the number of ABCC members is similar (9 members both in ABCCs and ABCHs). Interestingly, a recent analysis of the ABC complement of >150 arthropod species found that the H subfamily shows expansions in hemipterans [97]. This is consistent with the number of Triin-ABCH transcripts found in this work, where these sequences represent almost 30% of the entire family, although annotated genomic information

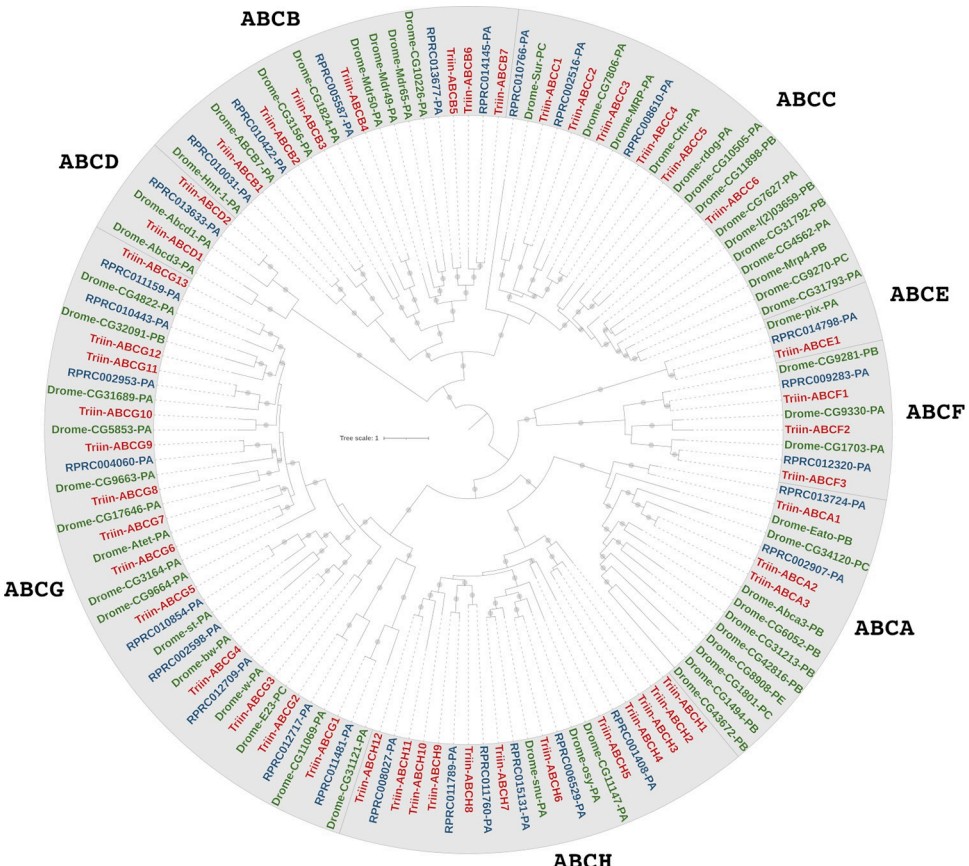

**Fig 6. Phylogenetic tree of the ABC transporter superfamily from *T. infestans*, *R. prolixus* and *D. melanogaster*.**
The maximum-likelihood tree was constructed using IQ-Tree and protein sequences obtained from *T. infestans*
transcriptome (Triin-, red, available in Table G in S2 File) and from *R. prolixus* (RPRC-, blue, VectorBase ID is shown)
and *D. melanogaster* (Drome-, green, FlyBase gene name and isoform are shown) genomes. Only those sequences
complete enough for phylogenetic characterization were included in the tree (remaining sequences were given a
preliminary classification shown in brackets in Table G in S2 File). *Triatoma infestans* sequences were numbered
according to their position in the tree. Branches with SH-aLRT support values > 80 are marked with a grey dot. The
tree was rooted at midpoint.

from the species is needed to rule out if any of them represent allelic variants or products of
alternative splicing. In mosquitoes, ABCH expression was enriched in sensory structures [99],
whereas in the hemipteran *Nezara viridula* the expression was enriched in the posterior
extreme of the intestine [97].

ABCB accounts for more than 10% of the sequences within the family in both triatomine
species (7 ABCB in *T. infestans* and 5 in *R. prolixus*, being HTs more abundant than FTs) and
the subfamilies A, D and E have the same number of transcripts in both triatomine species (3,
2 and 1 members respectively, Fig 6 and Table G in S2 File). Previous evidence indicates that
the expression of ABCA and ABCD tends to be higher in *An. gambiae* midgut, whereas ABCE
has an ubiquitous expression in mosquitoes [98].

The phylogenetic analysis shows a *T. infestans* orthologue for most *R. prolixus* sequences,
with the exception of one ABCC and two ABCG transporters (RPRC010766-PA,
RPRC002598-PA and RPRC010443-PA). Within the ABCH subfamily, a specific *T. infestans*
expansion is observed, with 4 transcripts including the upregulated Triin_ABCH2 (TRINI-
TY_DN3318_c1_g1_i9.p1, Fig 6 and Tables C and D in S2 File). Additional studies are needed

to address if this *T. infestans* ABCH expansion plays a role in the response to insecticides and/or resistance, as observed for members of other expanded subfamily of ABC transporters [102]. Although ABCH function is poorly studied, it has been suggested a possible role of these proteins in the response to xenobiotics due to their moderate similarity to ABCG family [103]. Moreover, the overexpression of ABCH after insecticide treatment was also observed in the coleopteran *T. castaneum* [104] and in two insecticide-resistant strains of *Plutella xylostella* [50], reinforcing the hypothesis of a role of these proteins in the tolerance and/or resistance to insecticides. Interestingly, functional analysis by RNAi both in *N. viridula* [97] and *T. castaneum* [101] suggested a role of ABCH members in cuticular formation.

The analysis of the *T. infestans* ABC superfamily expression profile shows that subfamilies E and F, along with some members of subfamily H have the highest expression in both conditions within the superfamily (Fig 7). There are also ABCs that show low expression in both conditions, especially some members of C, G and H subfamilies. With the exception of A, E and F subfamilies, the others show different levels of expression of its members: the most contrasting case is that of H subfamily, which has members that show very low expression in both conditions, while others are highly expressed (Fig 7). These differences could be due to the expression of members restricted to particular tissues, as discussed above.

## Concluding remarks

Here we present a *de novo* assembled transcriptome and the first quantitative RNA-Seq study performed on *T. infestans*, which is the main vector of Chagas disease in the Southern Cone. Four hours after the exposure to a sublethal dose of deltamethrin in a low-pyrethroid resistant population, an activation of the transcription machinery was observed, leading to significant changes in expression of genes belonging to ABC transporters, HSP70, CSP and OBP families. Transcripts related to transcription and translation processes, energetic metabolism and cuticle rearrangements were also modulated after intoxication. In a recent work, Ingham *et al.* [94], demonstrated that the response to a sub-lethal dose of a pyrethroid in the mosquito *An. coluzzii* elicits a transcriptomic response that can be divided in different temporal phases. We consider that an activation of detoxification-related gene families not observed in the analyzed population (such as CYPs, GSTs and CCEs) could also occur in a posterior phase of the temporal response to an intoxication with pyrethroids. Further experiments will be necessary to characterize the detoxificant response in this species on a temporal scale.

The modulation of CSPs expression is of particular interest, given that growing evidence is supporting the central involvement of this family in insecticide resistance, a fact that was overlooked until very recently. Even though the mechanisms by which CSPs contribute to insecticide tolerance and resistance remain to be elucidated, the sequestering and masking of xenobiotic molecules is the principal hypothesis to date [31,105]. It is expected that in the near future the field of insect detoxification will pose more attention to this gene family, and many open questions will start to be answered. This could lead to the identification of new chemical inhibitors for CSP that could be incorporated in insecticide formulations, in order to augment the susceptibility and/or revert resistance to pyrethroids by insect pests.

In the absence of an annotated genome, the transcriptomic data provided here constitute a key resource for molecular and physiological studies in *T. infestans*. This information will be necessary in further studies directed to particular genes, such as RNAi-mediated gene silencing, expression modulation in particular tissues and/or at different time points, as well as comparisons of gene expression in susceptible and resistant populations. The transcriptomic analysis of field populations with different resistance levels and/or different time points in the response to deltamethrin could reveal other gene families involved in the phenomenon. The

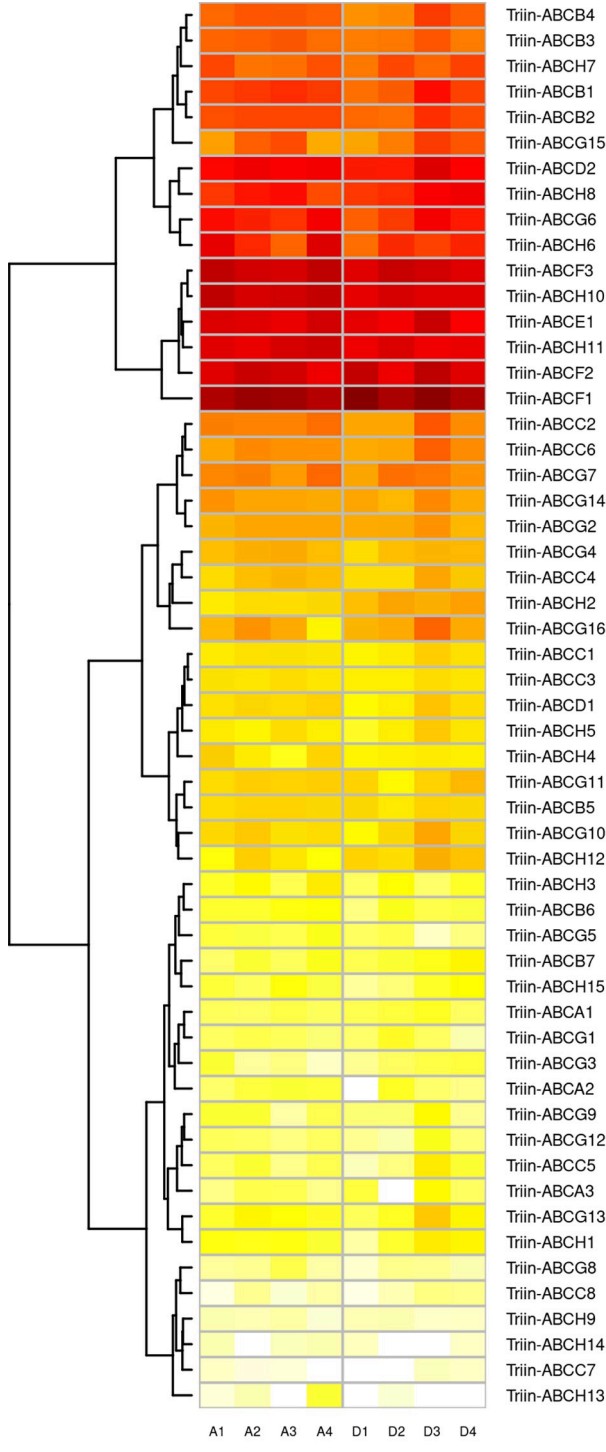

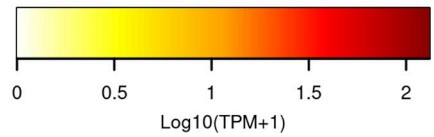

**Fig 7. Effect of deltamethrin on the transcription of *T. infestans* ABC transporters.** The heatmap was created using Transcript per Million (TPM) as input of the gplot R package. Transcript abundance was represented as Log$_{10}$(TPM +1) where white/red represents the lowest/highest expression. A dendrogram was plotted using a hierarchical clustering of gene expression values based on Euclidean distance and complete linkage method for clustering. A: acetone (control) samples. D: deltamethrin-treated samples.

results presented here, along with further studies on detoxification and insecticide resistance in *T. infestans*, will be useful for the rational design of an integrated vector control strategy.

## Supporting information

**S1 File. Summary of the pipeline used for data curation and analysis.**
(PDF)

**S2 File.** Table A: Sequencing results and number of reads after processing raw data. A: Acetone-treated samples (control). D: Deltamethrin-treated samples. Table B: Number of reads mapped to the non-redundant CDS database. Table C: Differentially expressed genes between acetone (control) and deltamethrin-treated samples. FC: Fold-change; FDR: False Discovery Rate. Table D: Sequence and putative annotation of transcripts with a significant change in their abundance after treatment with deltamethrin. For BLASTx searches, results are displayed as: Subject ID | Query (start-end): Start-end of alignment in query, Subject (start-end): Start-end of alignment in subject | % ID: Percentage of identical matches | evalue: Expect value. For SwissProt searches, subject complete name and taxonomic classification are also included. For HMMscan output (Pfam column), results are shown as: target accession | target name | target description | start-end of alignment | E: independent E-value. KEGG results are displayed as: KEGG Genes database (KEGG: organism code: gene identifier) and/or KEGG Orthology ID (KO). Gene ontology annotations from BLASTx and Pfam are displayed as: GO ID; GO category; GO name. Different results for the same sequence are shown separated by a slash. A dot in the cell indicates no results. Table E: *Triatoma infestans* CSP sequence analysis. Table F: *Triatoma infestans* and *R. prolixus* HSP70 sequence analysis. Table G: *Triatoma infestans* and *R. prolixus* ABC transporters sequence analysis. The classification indicated in brackets is given for sequences with <300 amino acids, NBD/TMD domains absent or mostly incomplete and/or fusioned proteins, based on a preliminary phylogenetic analysis and should be confirmed when complete sequences become available. The sequences painted in grey correspond to fragments of ABCs which were not classified and were excluded from the ABC repertoire of each species.
(XLSX)

**S1 Fig. Graphical summary of BUSCO assessment results of the *T. infestans* database of translated non-redundant CDS against Hemiptera.**
(TIF)

**S2 Fig. Sequence alignment of a fragment of domain II from the voltage-gated sodium channel.** S1: pyrethroid-susceptible; R3: pyrethroid-resistant from Güemes department [9]. The fragment of TRINITY_DN17507 shown here is represented by two sequences in the non-redundant CDS dataset. Nucleotide changes in comparison to S1 are shown in bold. Yellow: Codon where L925I substitution can occur. Red: Codon where L1014F substitution can occur (not present in any of the sequences analyzed here). Grey: Codon with a silent mutation detected in the assembled transcriptome, present in 63% of the mapped reads. Asterisk indicates conserved positions.
(TIF)

**S3 Fig. Volcano plot showing the effect of deltamethrin treatment on transcript abundance in *T. infestans*.** X-axis and y-axis represent log2 fold-change (FC) differences between the compared groups and statistical significance as the negative Log of P-values, respectively. The abundance of 9,887 transcripts (generated after the filtering step) was plotted. Transcripts with p-value (FDR) < 0.05 are indicated with red dots and non-significant transcripts are shown as black dots.
(TIF)

## Acknowledgments

The authors acknowledge Fausto Gonçalves and the Bioinformatics Platform of René Rachou Institute for providing the computational resources to perform the analysis and Dr. Brian Haas from Broad Institute for his kind assistance with the use of Trinity software. LNI and LIT are members of Argentinean Network for the Research in Pesticide Resistance (RAREP).

## Author Contributions

**Conceptualization:** Lucila Traverso, Gastón Mougabure Cueto, Sheila Ons.

**Data curation:** Lucila Traverso, Jose Manuel Latorre Estivalis.

**Formal analysis:** Lucila Traverso, Jose Manuel Latorre Estivalis.

**Funding acquisition:** Sheila Ons.

**Investigation:** Lucila Traverso, Sheila Ons.

**Methodology:** Lucila Traverso, Jose Manuel Latorre Estivalis, Gabriel da Rocha Fernandes, Georgina Fronza, Patricia Lobbia, Gastón Mougabure Cueto.

**Project administration:** Sheila Ons.

**Supervision:** Gabriel da Rocha Fernandes, Sheila Ons.

**Validation:** Lucila Traverso.

**Writing – original draft:** Lucila Traverso, Sheila Ons.

**Writing – review & editing:** Lucila Traverso, Jose Manuel Latorre Estivalis, Sheila Ons.

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
