## [Decision Letter · Decision Letter 0]

14 Mar 2022

Dear Dr Ons,

Thank you very much for submitting your manuscript "Transcriptomic modulation in response to an intoxication with deltamethrin in Triatoma infestans, vector of Chagas disease" for consideration at PLOS Neglected Tropical Diseases. As with all papers reviewed by the journal, your manuscript was reviewed by members of the editorial board and by several independent reviewers. In light of the reviews (below this email), we would like to invite the resubmission of a significantly-revised version that takes into account the reviewers' comments. 

I am sorry about the delay, but obtaining reviewers took more time than expected. The reviewers have identified some issues to be addressed, which I believe can be done without any additional experiments. I look forward to seeing the revision.

We cannot make any decision about publication until we have seen the revised manuscript and your response to the reviewers' comments. Your revised manuscript is also likely to be sent to reviewers for further evaluation.

Sincerely,

Joshua B. Benoit

Associate Editor

Jan Van Den Abbeele

Deputy Editor

I am sorry about the delay, but obtaining reviewers took more time than expected. The reviewers have identified some issues to be addressed, which I believe can be done without any additional experiments. I look forward to seeing the revision.

Reviewer's Responses to Questions

**Key Review Criteria Required for Acceptance?**

**Methods**

-Are the objectives of the study clearly articulated with a clear testable hypothesis stated?

-Is the study design appropriate to address the stated objectives?

-Is the population clearly described and appropriate for the hypothesis being tested?

-Is the sample size sufficient to ensure adequate power to address the hypothesis being tested?

-Were correct statistical analysis used to support conclusions?

-Are there concerns about ethical or regulatory requirements being met?

Reviewer #1: Methods are clearly described, as stated in the comments below, the non-standard pipelines should bee available on github or alternatively, the commands used reported in the manuscript.

Reviewer #2: The objectives of the study are clearly articulated. The study design is appropriate and were correctly analysed

No concerns about ethical or regulatory requirements.

Reviewer #3: The study design and the methods used for transcriptomic analysis are in general appropriate and correctly articulated with the objectives. However, I think there are some minor points the authors should address before the manuscript is acceptable for publication.

- As the authors recognize that the resistance phenomenon may be very variable depending on the resistance factor being involved, they should not generalise their findings to the whole species. Therefore, the title and other parts of the manuscript should reflect that the study is based on a particular low resistance T. infestans strain.

- Please, check the availability of the transcriptomic data, as searching within the NCBI database shows that the bioproject does not exist.

**Results**

-Does the analysis presented match the analysis plan?

-Are the results clearly and completely presented?

-Are the figures (Tables, Images) of sufficient quality for clarity?

Reviewer #1: The results are well described and interesting.

Figure 1 - I assume these are counts of some kind? Should specify what exactly this is.

Figure 2/3/6 - I can't read the branch labels at all

Some CSPs were also down regulated in An. gambiae after exposure

evolutive pressure should be evolution pressure

A little more details about the tissue specificity of the ABC family would be interesting for discussion

Reviewer #2: Results are clearly and completely presented.

Figures and tables are okay.

Reviewer #3: - In general, the results are clearly and exhaustively presented, especially those related to chemosensory proteins. However, the authors did not validate the differential expression results obtained from RNA-seq by any independent method. qPCR has historically been used to confirm data obtained in transcriptomics studies and despite the debate on whether qPCR validation of transcriptomics data is needed there is a consensus that it is especially useful to corroborate de expression of genes with low fold change values, which is the case of many genes in this study. Thus, I think the authors should carry out a qPCR analysis on some randomly chosen genes using independent cDNA samples of this Triatoma infestans strain in order to validate their present differential expression results. Furthermore, authors rose questions about the lack of differential expression in detoxifying genes such as cytochrome P450, esterases and others as overexpression of these genes has been reported in previous studies, which led them to speculate that this discrepancy might be related to insect strain being used and exposure time. And this issue can be easily solved with a qPCR analysis of genes coding for detoxifying enzymes in order to corroborate or not the results from RNA-seq.

- please include in Fig. 1 legends a brief explanation of what does the dendrogram on the left mean.

**Conclusions**

-Are the conclusions supported by the data presented?

-Are the limitations of analysis clearly described?

-Do the authors discuss how these data can be helpful to advance our understanding of the topic under study?

-Is public health relevance addressed?

Reviewer #1: See above

Reviewer #2: The conclusionsare supported by the data presented.

Reviewer #3: In general, the conclusions are supported by the data presented and the relevance of the genomic data is correctly discussed. May be the authors could discuss the limitations of the study related to the insect strain used for the analysis (why they chose to use a single low resistance strain instead of two or three strains with different resistance levels), the time the insects were processed after insecticide application (why they chose a 4 h exposure time instead of a longer one).

**Editorial and Data Presentation Modifications?**

Reviewer #1: Intro - minor corrections

The line ending in dissimilar success needs rephrasing

'its extension', I guess that should be 'its extent', also a citation would be good

The ABC superfamily, which is much less studied...have been associated...

Would be nice to be a bit more descriptive in the line starting 'we also performed a comprehensive...'

Reviewer #2: (No Response)

Reviewer #3: (No Response)

**Summary and General Comments**

Reviewer #1: This study utilises a triatoma bug with low levels of insecticide resistance from Argentina to characterise the impact of pyrethroid exposure on the transcriptome 4hours after application. The authors characterise chemosensory proteins and ABC transporters due to overexpression and prior role in resistance. Overall, the paper is very good, easy to follow and interesting. The authors do a very good job of explaining the results and linking them to published literature on pyrethroid resistance. I recommend this paper is publishable pending the described changes. I congratulate the authors on a nice paper of an understudied vector.

Is there a github with the .pl scripts used? It would also be beneficial to have the command used for the curation of the data as this doesn't follow a standard pipeline.

Further detail is needed about the resistance level, it is hard to know what 'low' resistance is - there needs to be some form of reference here. 

33321 transcripts seems like a very high number - how many of these are predicted to be splice variants of the same gene? I would expect somewhere in the region of 13k

Table 4 is really hard to follow - I know it is a lot of work, but it needs some kind of reformating to make it more readable. Some of the main points are lost.

Reviewer #2: Comments to the manuscript PNTD-D-21-01722 intended as an research article in PLOS Neglected Tropical Diseases entitled “Transcriptomic modulation in response to an intoxication with deltamethrin in Triatoma infestans, vector of Chagas disease” by Traverso L, Estivalis JML, Fernandes GR, Fronza G, Lobbia P, Cueto GM, Ons S. 

They present a very interesting manuscript investigating the transcriptomic alterations caused by the synthetic pyrethroid deltamethrin in a low pyrethroid-resistant population of Triatoma infestans. The methods used and the analyses performed seem appropriate and solid. The manuscript has a good and comprehensive introduction to the topic and the results are clear and concise presented. 

The work presented is to a large extend descriptive reporting of data, which is the nature of an initial transcriptome paper. The authors related their findings nicely to previous findings in other insects with pyrethroids or other insecticides. The authors should be praised for having a focus on e.g. chemosensory proteins and not just CYPs or GSTs, but it would be very interesting with some hypotheses or interesting questions on CSPs to enlighten the discussion and add more intellectual value to the manuscript.

The study is an important stepping stone to focus on the insecticide detoxification in Triatoma infestans.

Reviewer #3: The manuscript by Traverso et al. presents a transcriptomics analysis of Triatoma infestans first-stage nymphs from a low pyrethroid-resistant focus 4 hours after being exposed to a sublethal dose of deltamethrin. Whilst the selection of a single population from a region with a wide range of resistance levels and the short time of insecticide exposure may limit the findings of the study, relevant information on the differential expression of some genes has been obtained, especially regarding the relationship between chemosensory proteins and insecticide resistance. And most importantly, the study contributes to increasing the genomic information of this major vector of Chagas disease as its genome is not currently available.

PLOS authors have the option to publish the peer review history of their article (what does this mean?). If published, this will include your full peer review and any attached files.

Reviewer #1: Yes: Victoria A Ingham

Reviewer #2: No

Reviewer #3: No
---

## [Decision Letter · Decision Letter 1]

7 Jun 2022

Dear Dr Ons,

We are pleased to inform you that your manuscript 'Transcriptomic modulation in response to an intoxication with deltamethrin in a population of Triatoma infestans with low resistance to pyrethroids' has been provisionally accepted for publication in PLOS Neglected Tropical Diseases.

Best regards,

Joshua B. Benoit

Associate Editor

Jan Van Den Abbeele

Deputy Editor

Reviewer's Responses to Questions

**Key Review Criteria Required for Acceptance?**

**Methods**

-Are the objectives of the study clearly articulated with a clear testable hypothesis stated?

-Is the study design appropriate to address the stated objectives?

-Is the population clearly described and appropriate for the hypothesis being tested?

-Is the sample size sufficient to ensure adequate power to address the hypothesis being tested?

-Were correct statistical analysis used to support conclusions?

-Are there concerns about ethical or regulatory requirements being met?

Reviewer #1: The methods are well described.

Reviewer #2: The study design is appropriate and were correctly analysed.

Based on the arguments by the authors, it is found acceptable not to include qPCR data

Reviewer #3: (No Response)

**Results**

-Does the analysis presented match the analysis plan?

-Are the results clearly and completely presented?

-Are the figures (Tables, Images) of sufficient quality for clarity?

Reviewer #1: Very well explained

Reviewer #2: Results are clearly and completely presented. Figures and tables are okay.

Reviewer #3: (No Response)

**Conclusions**

-Are the conclusions supported by the data presented?

-Are the limitations of analysis clearly described?

-Do the authors discuss how these data can be helpful to advance our understanding of the topic under study?

-Is public health relevance addressed?

Reviewer #1: Yes, conclusions are fully supported by data

Reviewer #2: The conclusions are supported by the data presented.

Reviewer #3: (No Response)

**Editorial and Data Presentation Modifications?**

Reviewer #1: (No Response)

Reviewer #2: (No Response)

Reviewer #3: (No Response)

**Summary and General Comments**

Reviewer #1: The authors have answered all my concerns and I congratulate them on a lovely piece of work.

Reviewer #2: The authors have followed suggestions or answered comments and suggestions by the reviewers, which improved the manuscript.

Based on the arguments by the authors, it is found acceptable not to include qPCR data.

Reviewer #3: The authors have made sufficient amendments to the manuscript, transcriptomic data are currently available, thus the manuscript should now be accepted for publication. Congratulations to the authors for this very interesting research.

PLOS authors have the option to publish the peer review history of their article (what does this mean?). If published, this will include your full peer review and any attached files.

Reviewer #1: No

Reviewer #2: No

Reviewer #3: No

---

## [Editor Report · Acceptance letter]

24 Jun 2022

Dear Dr Ons,

We are delighted to inform you that your manuscript, "Transcriptomic modulation in response to an intoxication with deltamethrin in a population of *Triatoma infestans* with low resistance to pyrethroids," has been formally accepted for publication in PLOS Neglected Tropical Diseases.

Best regards,

Shaden Kamhawi

co-Editor-in-Chief

Paul Brindley

co-Editor-in-Chief
